# Investigation of Deoxidation Process of MoO_3_ Using Environmental TEM

**DOI:** 10.3390/ma15010056

**Published:** 2021-12-22

**Authors:** Peijie Ma, Ang Li, Lihua Wang, Kun Zheng

**Affiliations:** Beijing Key Laboratory of Microstructure and Properties of Solids, Faculty of Materials and Manufacturing, Beijing University of Technology, Beijing 100124, China; peijiema@126.com (P.M.); wlh@bjut.edu.cn (L.W.)

**Keywords:** TEM, in situ, deoxidation, MoO_3_

## Abstract

In situ environmental transmission electron microscope (ETEM) could provide intuitive and solid proof for the local structure and chemical evolution of materials under practical working conditions. In particular, coupled with atmosphere and thermal field, the behavior of nano catalysts could be directly observed during the catalytic reaction. Through the change of lattice structure, it can directly correlate the relationship between the structure, size and properties of materials in the nanoscale, and further directly and accurately, which is of great guiding value for the study of catalysis mechanism and the optimization of catalysts. As an outstanding catalytic material in the application of methane reforming, molybdenum oxide (MoO_3_)-based materials and its deoxidation process were studied by in situ ETEM method. The corresponding microstructures and components evolution were analyzed by diffraction, high-resolution transmission electron microscopy (HRTEM) and electron energy loss spectrum (EELS) techniques. MoO_3_ had a good directional deoxidation process accompanied with the process of nanoparticles crushing and regrowth in hydrogen (H_2_) and thermal field. However, in the absence of H_2_, the samples would exhibit different structural evolution.

## 1. Introduction

Molybdenum (Mo)-based materials are often used as efficient catalysts for various heterogeneous gas-solid catalytic reactions [1,2]. Primarily, molybdenum oxide (MoO_3_)/molybdenum carbide (MoC) catalysts are widely used in the chemical industry due to their value in the carbon cycle catalysis [3,4], including methane reforming [5]. As an excellent catalytic and substrate material, MoC also perfectly compounds other materials to achieve better results. For example, Ding Ma et al. employed α-MoC in hydrogenation reaction, showing high selectivity and yield because α-MoC and nitrogen compounds preferentially activate the C=O and C−OH bonds over C=C and C−C [6]. Interestingly, MoO_3_ is also the raw material used to produce MoC [7]. Xinhe Bao and colleagues prepared a highly reactive Au/MoC catalyst using MoO_3_ and Au nanoparticles. In the process, the MoO_3_ to MoC transition and the intermediate states, such as MoO_x_C_y_ and Au dispersion processes, were completed by the Strong Metal-Support Interaction (SMIS) [8]. The first step of the MoO_3_ deoxygenation process, both with MoO_3_/MoC catalysts and during the molybdenum carbide reaction, deserves to be explored in detail [9,10]. In addition, previous work pointed out the connection between deoxygenation sites and the active sites of the subsequent reaction, which helps explore the evolution of reaction sites [11]. Moreover, MoO_3_ and MoO_x_ are two-dimensional materials widely used in devices, which is one of the significances of exploring MoO_3_ deoxidation [12,13].

It is of great significance to explore the generations of catalysts and process of reaction [14,15]. And there are many in situ methods, such as X-ray diffraction patterns (XRD), Fourier Transform Infrared (FTIR), and X-ray Absorption Fine Structure (XAFS) are widely used in catalytic research, also the thermodynamic experiments mostly analyze the structure from a macro perspective rather than directly observing the reaction [14]. At present, there are few means for direct observation of catalytic reactions [16]. However, the recent development of environmental transmission electron microscope (ETEM) makes it possible to observe the nanomaterials’ performances during chemical reactions at an atomic scale [17,18]. In situ techniques such as thermal, electrical, and force fields are used to study the dynamics under external conditions during catalyst evolution and track the catalyst motion at the atomic scale to observe the atomic dynamics in real-time [19]. Therefore, the structure-activity relationships and reasons for activity could be better understood. In catalysis, many catalytic mechanisms are not clearly comprehended. The catalytic experiment can only show that some reactions are promoted, but the reason for their activity is still unknown [18]. Moreover, selective exploration should also get enough attention at the atomic scale, to guide more targeted, cheaper, and sustainable new catalysts [14].

In recent years, ETEM has been widely used to study heterogeneous catalytic reactions to effectively observe surface reactions, according to the high-resolution transmission electron microscopy (HRTEM) images attained [20,21]. Many areas are explored in situ TEM, such as the oxidation and reduction of materials [22,23] and the evolution of the catalyst in reactions, like ammonia synthesis [24,25]. HRTEM captures the detailed information of the catalyst changes, analyzes the phase changes of the sample with diffraction information, then provides insight into the material’s structure and correlates its performance accordingly [26]. On this basis, the analysis of elements and valence states combined with electron energy loss spectrum (EELS) or energy Dispersive X-Ray Spectroscopy (EDX) analysis is an excellent complement to electron microscopy [20]. Therefore, in situ TEM has irreplaceable significance in understanding the structure-activity relationship [27].

In this article, MoO_3_ deoxidizing process was studied in situ by ETEM, under thermal heating and a hydrogen (H_2_) atmosphere. The deoxidization process starts with the temperature increasing, and the structure breaks down simultaneously. The deoxidization process continues with the continuous temperature increase, and the broken structure grows again. In contrast, different phenomena are present under pure thermal heating in a vacuum. Specifically, there are no splitting and growth phenomena, but the MoO_3_ sublimation is evident. Simultaneously, the pressure and reaction location may be related to the active site and catalytic activity. These phenomena were verified by electron diffraction maps and EELS plots. The deoxygenation process of MoO_3_ is carried out on the atomic scale, which is important for understanding and exploring the production process of MoC. Also, it is important to study the relationship between the evolution and structural properties of Mo-based catalysts in the catalytic process.

## 2. Results

### 2.1. Deoxidation of MoO_3_

The MoO_3_ used in this study is the commercially available MoO_3_ nanoparticles (NPs) with a 50–100 nm diameter dispersed in ethanol solution and cast on a SiNx-based heating chip. The NPs were loaded into the ETEM by an in situ heating holder (Wildfire S3, DENS solutions). For the in situ experiment, the temperature was controlled by a DENs heating system, and the atmosphere was controlled with an ETEM gas path system.

#### 2.1.1. Characteristics Analysis of Molybdenum Oxide MoO_3_

Before the in situ experiment, the samples were preliminarily analyzed, and to obtain clearer images for the study, thinner NPs were chosen. Figure 1a shows a low magnification TEM image for MoO_3_ NPs dispersed on the carbon film, where the NPs size is uniform. Figure 1b is the spectrum of the sample collected using EELS, corresponding to the MoO_3_position. The Mo-M_4,5_ and M_2,3_ edges are attributed to Mo, and the O-K edge corresponds to O. Generally, the O/Mo ratio is obtained through quantitative analysis of the Mo-M_3_ and O_k_ edges [16]. Therefore, in the following in situ experiments, the variation characteristics were acquired in real-time with the structural analysis. The XRD data in Appendix A shows that the main peak positions accurately correspond to each crystal plane. Apart from the spectroscopic analysis of MoO_3_ samples, static electron microscopy characterization was performed, and the structure and phase of samples were analyzed. Figure 1c shows the HRTEM images and diffraction patterns from the [10] direction of orthorhombic MoO_3_ samples. Figure 1d is the enlarged red rectangular area in Figure 1c. The (100) and (001) spots in the diffraction pattern in Figure 1c correspond to the crystal plane with 3.64 nm 3.92 nm spacings in Figure 1d, respectively [12]. According to TEM images analysis, the crystal direction of the dispersed sample is mostly displayed in the [10] direction.

#### 2.1.2. MoO_3_Structural Changes in H_2_ Atmosphere

Figure 2a shows the experimental schematic of the in situ ETEM and heated scaffold system. The ventilation function of the environmental in situ ETEM introduces H_2_ gaseous spheres as an external field. Meanwhile, the heating support system controls the temperature to achieve the heating of the sample. As shown in Figure 2a, the NPs dispersed on the silicon nitride chip and heated. Under these conditions, real-time information of samples, including HRTEM photos and EELS mapping data, was collected.

Starting from MoO_3_ NPs, the MoO_3_ response when the conditions changed from reduction to redox was studied. H_2_ pressures of 0.1 mbar and 0.6 mbar were applied, and the temperature was gradually increased from 25 °C to 900 °C. Previous studies have shown that pressure affects the phase transition at different temperatures [9,28]. Figure 2b–g shows the MoO_3_ evolution in a 0.6 mbar H_2_ atmosphere. Moire fringes were generated near the characteristic temperatures, 250 °C and 800 °C, around the diffraction temperature. However, it is not evident at other temperatures like 600 °C. In other words, the temperature at which Moire fringes are produced is the temperature at which the phase transition takes place violently. On the other hand, it is the temperature at which the particle’s shape changes dramatically at later heating especially after 600 °C. The Moire fringes are the sample splitting caused by temperature change and the particle dislocation caused by deoxidation and successive phase transition. However, during stability periods, such as 400–600 °C, the Moire fringes are not prominent. Therefore, it was considered the stabilizing period for MoO_2_. At this point, the deoxidation process continues, but the phase transition is not significant and is not a major concern. Combined with diffraction information at different temperatures, the phase transition conclusion is consistent with the above analysis.

In addition to the study of Moire fringes, this work mainly demonstrates the following: (1) The splitting and growth of NPs reflect the shape changes; (2) Changes in oxygen content cause Deoxidation resulting in structural and compositional changes; (3) The phase change is mainly reflected by diffraction; (4) The relationship between position and non-synchronous reaction. The following sections will discuss the detailed analysis of these aspects in detail.

#### 2.1.3. Morphology and Phase Evolution during Deoxidation

The splitting and growth of the samples were revealed, and phase transition was analyzed by diffraction patterns and HRTEM. The images in Figure 2b–g show the splitting and growth process for a MoO_3_ NP. As the temperature increases to 450 °C, the overall shape of the NP splits into small fragments of ~10 nanometers. As the temperature rises to 550 °C, the NP regrows into 50 nm (Figure 3j) like the original structure before heating. Figure 3i shows the overall morphology of the reaction of the ~50 nm NP (the splitting and growth trends are consistent whether the size of NPs is different). From the changes in Figure 3a–h, the Moire fringe analysis in Figure 2 is also applicable under 0.1 mbar, except that the reaction temperature changes due to air pressure, and the overall trend remains unchanged. The fracture analysis may be driven by temperature or caused by structural recombination after the loss of oxygen.

The diffraction pattern at 400 °C, shows that the nanoparticle gradually changed into a mixed state of MoO_3_ and MoO_2_. When it reached 600 °C, MoO_2_ was present. At 800 °C, the presence of Mo simple substance could be confirmed, combined with the thermogravimetric experiment and the diffraction pattern analysis [29]. The Figure 3h and Appendix A indicates the diffraction spots corresponding to MoO_3_, MoO_2_, and Mo materials, and Appendix A proved the location of Mo by filtering analysis. The following comparative experiments were performed to discover whether the above changes are related to the H_2_ atmosphere or mainly dominated by temperature.

### 2.2. Compositional Evolution of MoO_3_

To verify the effect of the atmosphere on MoO_3_ NPs size change, in situ control experiments with heating from 25 °C to 900 °C were performed in a vacuum (Figure 4).

#### 2.2.1. Non-Deoxidation Sublimation of MoO_3_

Firstly, simultaneous splitting behavior and a large amount of sublimation occurred [30,31], and the molybdenum oxide sample sublimated at 600 °C. However, the splitting is irreversible, unlike the deoxidation process under hydrogen atmospheres. In addition, the diffraction pattern does not change, proving that the orthonormal MoO_3_ is stable, and no deoxidation and phase transformation has taken place. However, sublimation greatly reduces the quality of the nanoparticle. Therefore, the splitting of the nanoparticle is directly related to temperature. To the contrary, phase transition and regrowth are mainly caused by hydrogen-induced deoxidation, which is expected by the controlled experimental design.

#### 2.2.2. Composition Changes under Different Conditions

Combined with the analyses in Figure 2, Figure 3 and Figure 4, statistical analysis on the element proportion in sample evolution under the three conditions was conducts, as shown in Figure 4f. Hydrogen is the leading cause of deoxidation, and temperature is the condition to promote the reaction. Furthermore, a difference in air pressure will cause a change in the temperature at which the reaction occurs.

### 2.3. Evolution and Characterization of the Position

Figure 5 shows the structure, phase, and composition related to location evolution. Figure 5a is the TEM analysis of MoO_3_ NPs in the deoxidation process, and Figure 5b–g is the EELS mapping evolution in the red rectangular area at different temperatures. The EELS mapping gives a visual representation of the elemental processes, where the evolution of the oxygen content is evident, confirming the different oxidation states at different temperatures during the deoxygenation process. However, this process is accompanied by sublimation, as seen from the Mo region reduction, although the sublimation process under hydrogen is weak for pure heating (Figure 4).

Different phenomena will occur at different locations for the deoxidation reaction sample under the same conditions. The deoxidation processes at the edge (position 1) and inside (position 2) of the sample are asynchronous, where the deoxidation in the interior is much slower than that in the edge, shown by the EELS Mapping. In Figure 5, the deoxidation process at position 1 is performed rapidly compared with that at position 2. The reaction is violent because the edge is exposed first to theH_2_ atmosphere, and its thickness is relatively thin. The contact zone affects the internal deoxidation process and is related to the complex phase transition and mechanical structure changes. Notably, the deoxidation site is associated with the active site of the catalytic process [11], which needs to be further explored.

## 3. Discussion

The H_2_ pressure impacts the temperature at which the morphology changes, reflected by the splitting and growth of NPs, take place, as shown in Figure 2 and Figure 3 under different air pressures. This agrees well with the result of other studies [32]. The structure and composition changes caused by the oxygen content induced by the deoxidation were mainly analyzed by EELS. Temperature causes sublimation and splitting behavior. H_2_ atmosphere is the main factor determining the deoxidation conditions and the driving force of subsequent growth. When the reaction is at the nanometer scale, subtle position changes will also bring different reaction conditions, such as different deoxidation phenomena occurring at different positions with an area of a few nanometers. This behavior cannot be noticed in the macroscopic experiment, even if other means of exploration cannot directly explore the local phenomenon. The study of active sites is also an important aspect of catalytic reaction. In situ TEM could continue a complete experiment for MoC/MoO_3_ catalysts using methane reforming [33]. The detailed analysis of these aspects, which we will study in future research, is very important for studying the active sites of catalysts. The real-time information brought by in situ electron microscopy allows us to observe the direction of sample evolution in real-time during the experiment and directly corresponds to the nano-scale visual images to explain the structure-activity relationship.

## 4. Conclusions

In this work, in situ environmental TEM was used to reveal the deoxidation process of MoO_3_ at atomic scale in terms of structure and composition. The results show the sample underwent a deoxidation process with the increase of temperature in the H_2_ atmosphere. While in the absence of H_2_, MoO_3_ underwent sublimation only. In addition, it is also found that the H_2_ atmosphere in difference pressure leads to different deoxidation and phase transformation rates, suggesting that H_2_ is the main inducing condition for deoxidation. Besides, the deoxidation process is different at different locations of the same NPs; the edges tend to show more active reactions than the inside of the sample. The results have significant guidance for understanding the evolution process of molybdenum base materials and the relationship between the properties and structure of materials. The local evolution of samples is explored from a broader perspective, which provides a reliable experimental means for the future study of gas-solid catalysis, especially catalyst behavior.

## Figures and Tables

**Figure 1 materials-15-00056-f001:**
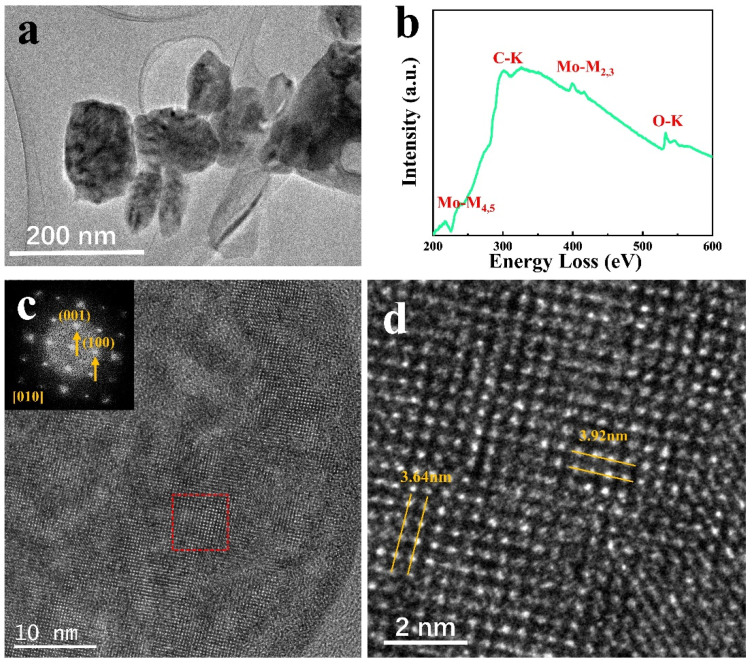
(**a**) TEM images of MoO_3_; (**b**) EELS of MoO_3_; (**c**,**d**) HRTEM images of MoO_3_.

**Figure 2 materials-15-00056-f002:**
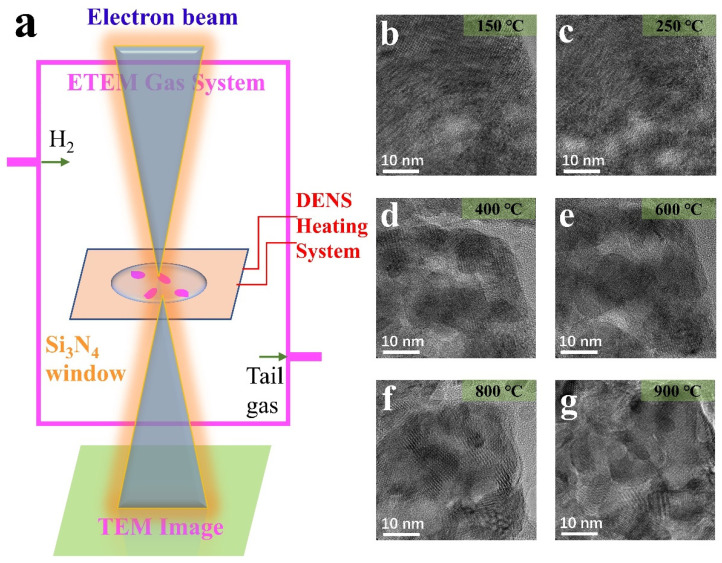
(**a**) Diagram of the experimental installation; (**b**–**g**) HRTEM images of the MoO_3_ NP by heating in H_2_ (0.6 mbar) at different temperatures.

**Figure 3 materials-15-00056-f003:**
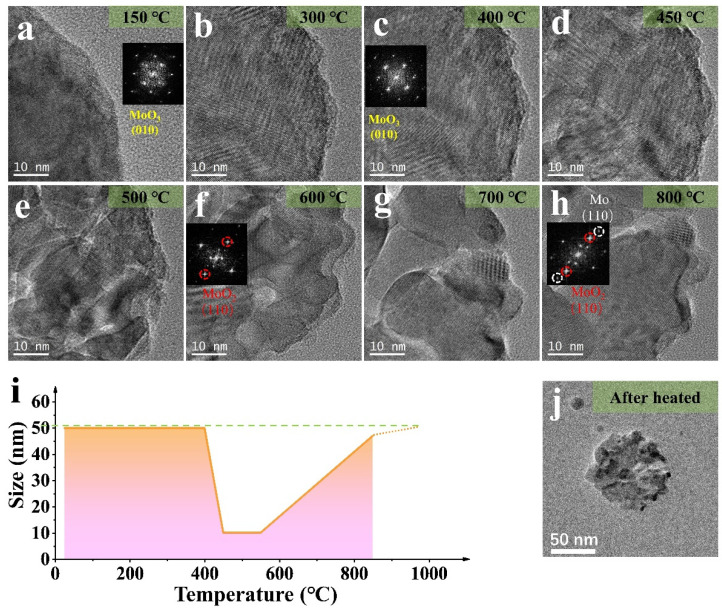
(**a**–**h**) HRTEM images of the MoO_3_ NP (~50 nm) by heating in H_2_ (0.1 mbar) at different temperatures (in Figure 3a,c,f,h, the corresponding stable diffraction images and label are given respectively); (**i**) The size change of the NP in (**a**–**h**); (**j**) The TEM image of NP after heated in H_2_ (0.1 mbar).

**Figure 4 materials-15-00056-f004:**
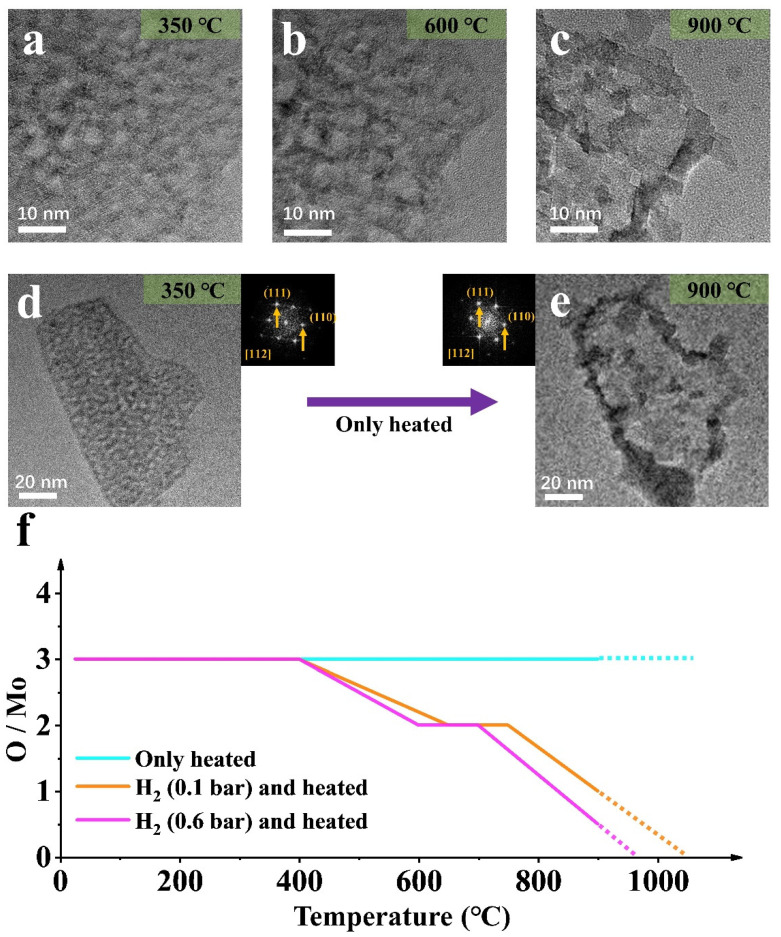
(**a**–**e**) TEM images of the MoO_3_ NP by only heating at different temperatures (in (**d**,**e**), the corresponding stable diffraction images and label are given respectively); (**f**) The element scale diagram changes of the NP in (**a**–**e**) at different temperatures.

**Figure 5 materials-15-00056-f005:**
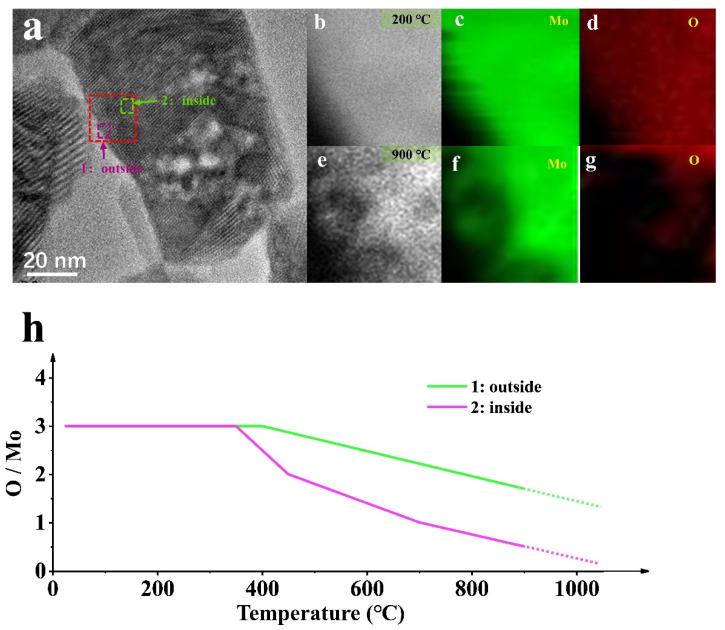
(**a**) HRTEM images of the MoO_3_ NP by heating in H_2_ (0.6 mbar); (**b**–**d**) The EELS mapping of NP representing the location of the red dotted line in (**a**) at 200 °C; (**e**–**g**) The EELS mapping of NP representing the location of the red dotted line in (**a**) at 900 °C; (**h**) The element scale diagram changes of the NP in (**a**) at different temperatures.

## Data Availability

No new data were created of analyzed in this study. Data sharing is not applicable to this article.

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
