# Peer review of "Investigation of Deoxidation Process of MoO_3_ Using Environmental TEM"

_materials, 2021, doi:10.3390/ma15010056_

Round 1
Reviewer 1 Report
Check line 29 “materals…”
Check line 52 “Thermal…”
Can the authors check the sentence at 69-70?
The description in the text of Figure 1 is not correct or does not correspond to the images in Figure 1, check Lines 91-95
From where does the C signal in the EELS spectra from MoO3 comes from?
Can the authors index the rest of the peaks in the XRD pattern shown in Figure S1?
Check “pressure” and “of” in line 120
Check “was” in line 124
Seems like Moire fringes are present all the time, can the authors comment on that? For example in figure 2e (600 °C) at the bottom right some Moire fringes can be appreciated, similarly in figure 2g (900 °C), how these observations fit in the discussion proposed?
Line 143: I would suggest to use “morphology” instead of “Appearance”
Figure 3 (i) should be "average size"? Or is it the size of one particle?
Line 160: authors meant 80 °C or 800 °C ? also check sentence “was possible exist”
Is there any structural evidence of Mo to exist at that temperature apart from the thermogravimetric experiment? Although a diffraction pattern it's mentioned however it is not shown.
Can the authors label the FFTs with “MoO3” shown in Figure 4 so it is more clear the message?
Is it Line 189 seems out of place?
The Supplementary materials has additional data that doesn’t seem to correspond to the present study, please check from figure S2 on.
Reviewer 2 Report
The paper is interesting and appropriate for Materials journal as it deals with a study of a Mo oxide by environmental TEM. However, the english language needs some revisions. The text is understadable, however, there are many errors dealing with singular/plural and grammar. I suggest authors to ask the help of a native english speaker or someone with equivalent skills to correct the language. Some examples are:
lines 10/11, working condition, should be, working conditions.
line 11, the behaviours, should be, the bahaviour
line 24, for varies of, should be, for a variety of
lines 32, MoO3 is also the raw materials to produce, should be, MoO3 is also the raw material used to produce
line 41, Move over., should be, Moreover,
line 43, the generation of catalyst, should be, the generation of catalysts
line 44, By contrast, should be, In contrast,
etc, etc,
Reviewer 3 Report
This paper deals with the study of deoxidation process of MoO3 using environmental TEM. The manuscript is quite well organized and report substantial new information on the subject. Only some marginal aspects should be revised.
In particular, the authors should take into consideration the following points:
-The English form should be revised in all the manuscript, presumably by an English mother tongue.
-The authors should specify the code used in the abstract (as for example EELS) to facilitate the reading.
-Please check and correct the typos. For example, “Move over” in the line 41, the capital letter of “Verified” in the line 72, etc.
-The authors should better explain or even better add some examples to how it is possible to use this technique in situ during a catalytic reaction.
Round 2
Reviewer 1 Report
The authors have presented an upgraded version of the manuscript according to the revision.